# Visualization of ferroaxial domains in an order-disorder type ferroaxial crystal

T. Hayashida[1], Y. Uemura [2], K. Kimura[1], S. Matsuoka [2], D. Morikawa [3], S. Hirose [4], K. Tsuda[5], T. Hasegawa [2] & T. Kimura [1✉]

Ferroaxial materials that exhibit spontaneous ordering of a rotational structural distortion with an axial vector symmetry have gained growing interest, motivated by recent extensive studies on ferroic materials. As in conventional ferroics (e.g., ferroelectrics and ferromagnetics), domain states will be present in the ferroaxial materials. However, the observation of ferroaxial domains is non-trivial due to the nature of the order parameter, which is invariant under both time-reversal and space-inversion operations. Here we propose that $NiTiO_3$ is an order-disorder type ferroaxial material, and spatially resolve its ferroaxial domains by using linear electrogyration effect: optical rotation in proportion to an applied electric field. To detect small signals of electrogyration (order of $10^{-5}$ deg $V^{-1}$), we adopt a recently developed difference image-sensing technique. Furthermore, the ferroaxial domains are confirmed on nano-scale spatial resolution with a combined use of scanning transmission electron microscopy and convergent-beam electron diffraction. Our success of the domain visualization will promote the study of ferroaxial materials as a new ferroic state of matter.

[1] Department of Advanced Materials Science, University of Tokyo, Kashiwa, Chiba 277-8561, Japan. [2] Department of Applied Physics, University of Tokyo, Tokyo 113-8656, Japan. [3] Institute of Multidisciplinary Research for Advanced Materials, Tohoku University, 2-1-1, Katahira,Aoba-ku, Sendai 980-8577, Japan. [4] Murata Manufacturing Co., Ltd., Nagaokakyo–shi, Kyoto 617–8555, Japan. [5] Frontier Research Institute for Interdisciplinary Sciences Tohoku University, 6-3, Aramaki Aoba, Aoba-ku, Sendai 980-8578, Japan. ✉email: tkimura@edu.k.u-tokyo.ac.jp

The order parameter characterizing ferroaxial materials is a rotational electric-dipole arrangement[1] and represented by a ferroaxial moment (or ferro-rotation moment) $\mathbf{A}$ defined as $\mathbf{A} \propto \sum_i \mathbf{r}_i \times \mathbf{p}_i$, where $\mathbf{r}_i$ denotes a position vector of electric dipole $\mathbf{p}_i$ from the symmetrical center of a structural unit[2,3]. For example, $\mathbf{A}$ is generated by head-to-tail arrangements of electric dipoles as illustrated in Fig. 1a. The $\mathbf{A}$ is an axial vector invariant under both time-reversal and spatial-inversion operations though other symmetries such as a mirror parallel to $\mathbf{A}$ is broken. The ferroaxial order is closely related to various phenomena including magnetoelectric couplings in multiferroics[4–6] and polar vortices in nanostructured materials[2,7]. Such an order is sometimes called ferro-rotational order[3,8], and these terms are often used to describe the existence of rotational distortions inducing finite $\mathbf{A}$ with or without a phase transition[4–6].

The symmetry aspect of ferroaxial transitions is detailed in ref. [9]. Among the 32 crystallographic point groups, there are 13 pyroaxial groups $(1, \bar{1}, 2, m, 2/m, 3, \bar{3}, 4, \bar{4}, 4/m, 6, \bar{6},$ and $6/m)$ which allow for finite $\mathbf{A}$. (The term "pyroaxial" is considered an analogue to the term "pyroelectric".) In all the pyroaxial groups except for $\bar{1}$, $\mathbf{A}$ is parallel to the principal axis of a crystal. Furthermore, among the 212 species that classify symmetry reductions in phase transitions, there are eight pure ferroaxial species that accompany neither ferroelectric, ferroelastic, nor gyrotropic transitions $(\bar{4}2m \to \bar{4}, 4/mmm \to 4/m, 4/mmm \to \bar{4}, 3m \to \bar{3}, \bar{6}2m \to \bar{6}, 6/mmm \to 6/m, 6/mmm \to \bar{6},$ and $6/mmm \to \bar{3})$[9,10]. In such pure ferroaxial transitions, we expect that a pair of domains with the opposite signs of $\mathbf{A}$ are formed (compare the left and right panels of Fig. 1a). A representative of such ferroaxial materials is multiferroic RbFe(MoO$_4$)$_2$ showing one of the eight pure ferroaxial transitions $(3m \to \bar{3})$[11]. Recently Jin and coworkers experimentally demonstrated the presence of domain states in this material by using rotational anisotropy second-harmonic generation (SHG) and showed the temperature variation of the areal ratio of the ferroaxial domains[8]. However, spatial distributions of ferroaxial domains have never been directly observed through the nature inherent in ferroaxial order. In this article, for the first time, we visualize ferroaxial domains of a newly proposed order–disorder type ferroaxial material NiTiO$_3$ in two different ways: scanning transmission electron microscopy (STEM) combined with convergent-beam electron diffraction (CBED) and linear optical microscopy via the linear electrogyration (EG) effect whose conceptual diagram is depicted in Fig. 1b. The former possesses a sensitivity to picometer-scale atomic displacements and a nanometer-scale spatial resolution[12], and therefore allows the direct observation of the rotational structural distortion with a high spatial resolution. Meanwhile, the latter has only a micrometer-scale spatial resolution but can visualize global domain structures.

## Results

**Order–disorder type ferroaxial transition in NiTiO$_3$.** Ferroelectric phase transitions are known to be classified into mainly two types: displacive type and order–disorder type. In the same manner, ferroaxial transitions will also be classified into these two types. RbFe(MoO$_4$)$_2$, that is, the only material in which a ferroaxial transition is studied, exhibits a ferro-rotational distortion mainly ascribed to displacements of oxygen atoms[11], and therefore can be said to be a displacive type ferroaxial crystal. In this study, we propose that a structural phase transition reported in NiTiO$_3$[13] is regarded as an order–disorder type ferroaxial transition (Fig. 2). At temperatures above $T_c \approx 1560$ K, the crystal structure of NiTiO$_3$ is described by the corundum structure (space group $R\bar{3}c$) which is envisage as a hexagonal close packing of the oxygen ions with Ni$^{2+}$ and Ti$^{4+}$ cations randomly occupying 2/3 of the octahedral interstices (Fig. 2a). With lowering temperature, cation ordering takes place at $T_c$ and results in a structural phase transition into the ilmenite structure (space group $R\bar{3}$). The low-temperature structure is characterized by an alternating sequence of Ni$^{2+}$ and Ti$^{4+}$ along the stacking direction of the closed-packed layers (Fig. 2b). Depending on the stacking sequence (Ni-Ti-Ni-Ti- or Ti-Ni-Ti-Ni-), two possible domain states develop at temperatures below $T_c$ (Fig. 2b, c). The transformation from the point group $\bar{3}m$ into $\bar{3}$ in NiTiO$_3$ is the same as that in RbFe(MoO$_4$)$_2$ and is nothing less than a ferroaxial transition. Indeed, as seen in Fig. 2d, e which depict two specific Ti ions and six oxygen ions bonded to these Ti ions, the direction of rotational distortions of oxygen ions (red arrows), i.e., the sign of $\mathbf{A}(\|\mathbf{c})$, is opposite in these two domain states (hereinafter, referred to as A+ domain and A− domain). These two domain states are related to each other by the operations whose symmetries are lost at the ferroaxial transition {e.g., two-fold rotation about [110] and $c$-glide operation with glide plane $\|$ (110)}. NiTiO$_3$ crystals used in this study were grown by the floating zone method (Methods). In the growth process, the specimens were once heated at temperatures above $T_c$ and then cooled down to room temperature, meaning that the crystals underwent the ferroaxial transition and are expected to consist of both A+ and A− domains.

**Identification of ferroaxial domains by STEM-CBED measurement.** The coexistence of a pair of ferroaxial domains

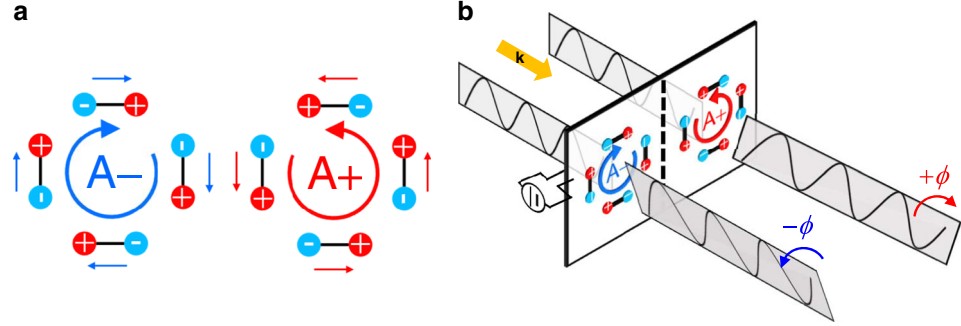

**Fig. 1 Ferroaxial order and linear electrogyration induced by ferroaxial order. a** Ferroaxial moment defined as $\mathbf{A} \propto \sum_i \mathbf{r}_i \times \mathbf{p}_i$, which characterizes ferroaxial materials. Here $\mathbf{r}_i$ denotes a position vector of electric dipole $\mathbf{p}_i$ from the symmetrical center of a structural unit. The sign of $\mathbf{A}$ is characterized by the rotation direction of the electric dipoles. A pair of ferroaxial domains with the opposite signs of $\mathbf{A}$ (A+ and A− domains) are illustrated. **b** When linearly polarized light propagates through a ferroaxial crystal, optical rotation is induced by the application of an electric field via linear electrogyration effect. Here $\mathbf{k}$ is the propagation vector of the light. The sign of rotation angle $\phi$ depends on that of $\mathbf{A}$, which allows the visualization of ferroaxial domains through the linear electrogyration effect.

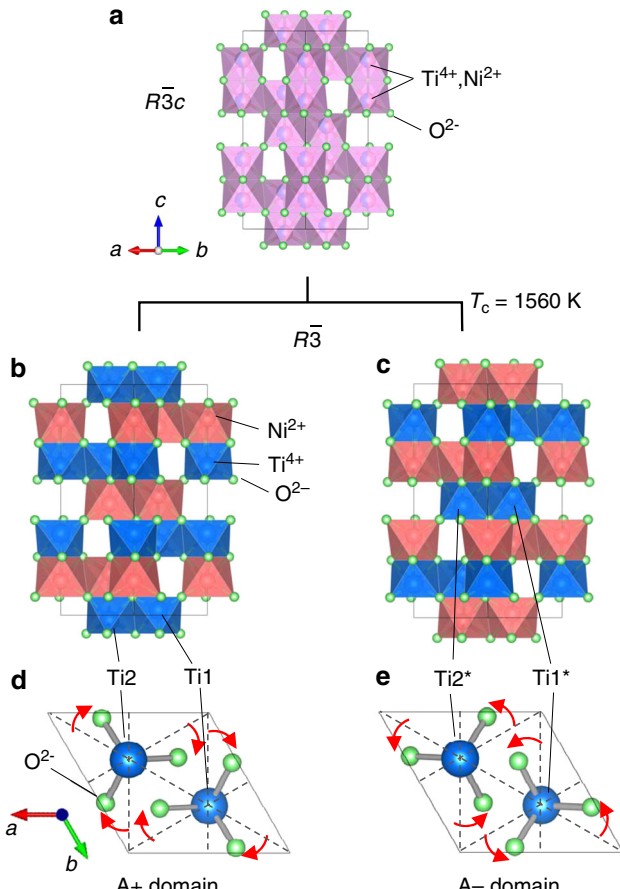

**Fig. 2 Order-disorder phase transition and formation of ferroaxial domains in NiTiO₃. a** The crystal structures of NiTiO₃ **a** above and **b**, **c** below the ferroaxial transition temperature ($T_c \approx 1560$ K). The crystal structure above $T_c$ belongs to a non-ferroaxial space group ($R\bar{3}c$) and is described by a disordered corundum structure with a random distribution of $Ni^{2+}$ and $Ti^{4+}$ ions at cation sites. With lowering temperature, cation ordering takes place at $T_c$ and results in a structural phase transition into an ordered ilmenite structure with a ferroaxial space group ($R\bar{3}$). Below $T_c$, thus, a pair of ferroaxial domain states with the opposite rotation direction, i.e., the opposite sign of axial vector **A**, are present (A+ and A− domains). **d**, **e** The c-axis views of the ferroaxial domains. Only two Ti ions [Ti1$^{(*)}$ and Ti2$^{(*)}$] and six oxygen ions are depicted. These ions form two TiO₃ triangular pyramids which are related by the space inversion with the inversion center at the midpoint between Ti1 and Ti2 ions. Red arrows denote the direction of rotational displacements of oxygen ions from the (110)-type planes (dotted lines) that correspond to the average oxygen positions between A+ and A−.

(A+ domain and A− domain) in a piece of the NiTiO₃ crystal was examined by the combined use of STEM and CBED[14]. This technique (hereinafter, referred to as STEM-CBED technique) possesses a sensitivity to picometer-scale atomic displacements and a nanometer-scale spatial resolution[12], and therefore allows us to visualize spatial distributions of various nanostructures such as polar nanostructures in ferroelectrics[15,16]. In the present study, we apply this technique to the observation of ferroaxial domains in NiTiO₃. The measurement details are described in Methods and Supplementary Note 1. Figure 3a shows a bright-field (BF)-STEM image obtained with the 001 incidences, and its magnified view of the area surrounded by a yellow framed box is displayed in Fig. 3b. Figure 3c, d show CBED patterns obtained at positions C and D in Fig. 3b, respectively, with the 001 incidences. Zeroth-order Laue zone (ZOLZ) reflections are seen near the center while

ring-shaped higher-order Laue zone (HOLZ) reflections on the fringe of the CBED patterns. Yellow arrowheads point characteristically intense HOLZ reflections, indicating that the CBED patterns of Fig. 3c, d are almost in a mirror image relation whose mirror plane is parallel to (110). Note that such a mirror operation is one of the symmetry elements which are present in the high-temperature $\bar{3}m$ phase but lost in the low-temperature $\bar{3}$ phase.

We performed computer simulations of the CBED patterns for the structure models of NiTiO₃ (see "Methods" and Supplementary Note 1). Figure 3e, f shows simulated CBED patterns with the [001] and [00$\bar{1}$] incidence, respectively, for the A+ domain. These two incidence configurations are converted into each other by the two-fold rotation about [110], and are equivalent to the measurements of a pair of ferroaxial domains (A+ and A− domains). The specimen thickness used for the simulations was 35 nm. The simulated CBED patterns in Fig. 3e, f are also in a mirror image relation reflecting the atomic arrangements in the two domains and well match up with the measured CBED patterns shown in Fig. 3c, d, respectively [compare the HOLZ reflections indicated by yellow arrowheads in Fig. 3c–f]. This result shows that ferroaxial domains with opposite signs of **A** are located at positions C (A+ domain) and D (A− domain) in Fig. 3b. In Fig. 3g, h, furthermore, we display STEM-CBED maps using the intensities of the HOLZ reflections at G and H (yellow-dotted circles in Fig. 3c, d), respectively. These maps clearly show spatial distributions of the intensity, meaning the formation of ferroaxial domains in the specimen used in this study. The location of domain boundaries in the entire sample area displayed in Fig. 3a was examined by observing the CBED patterns at various sample positions. As a result, only one flat boundary was revealed in the area (white dotted line in Fig. 3a). The crystal orientations of the A+ and A− domains separated by the boundary were determined from the CBED patterns and are schematically illustrated in Fig. 3i. The domain boundary is oriented parallel to the (110) plane. Thus, the coexistence of a pair of ferroaxial domains (Fig. 2d, e) is confirmed in terms of the structural characterization using the STEM-CBED technique on nanometer-scale spatial resolution.

**EG as a tool to observe ferroaxial domains.** Here we discuss another approach for observing ferroaxial domains. That is the approach by using the EG effect, i.e., optical rotation induced by an external electric field (see Fig. 1b). The EG effect was firstly described by Aizu[17] and Zheludev[18] independently in 1963-1964, and demonstrated in quartz crystals by Vlokh[19] in 1970, a half century ago. To date, this effect has been measured in various crystals[20,21] including PbWO₄[22] and Pb₅Ge₃O₁₁[23]. The EG effect is described by the change in the gyration tensor $g_{ij}$ as a function of an applied electric field $E$ and expressed as a power series,

$$g_{ij} = g_{ij}^{(0)} + \gamma_{ijk}E_k + \beta_{ijkl}E_kE_l + \cdots. \quad (1)$$

Here $g_{ij}^{(0)}$ represents natural optical rotation, and $\gamma_{ijk}(\beta_{ijkl})$ represents the linear (quadratic) EG effect. Hereinafter, the $z$ axis (the third axis) is taken as the principal axis. The linear EG effect characterized by the third-rank axial tensor $\gamma_{ijk}$ is possible in all point groups except for $m3m$, $\bar{4}3m$, and 432, while the quadratic one by the fourth-rank axial tensor $\beta_{ijkl}$ is only in noncentrosymmetric point groups. Note that, in centrosymmetric pyroaxial groups ($\bar{1}, 2/m, \bar{3}, 4/m$, and $6/m$), the natural optical rotation is absent. Furthermore, the Pockels effect and the inverse piezoelectric effect are not allowed, and therefore the linear EG effect is the only optical effect proportional to $E$. Considering these symmetry requirements, it can be said that the centrosymmetric pyroaxial crystals are ideal playgrounds to examine the linear EG

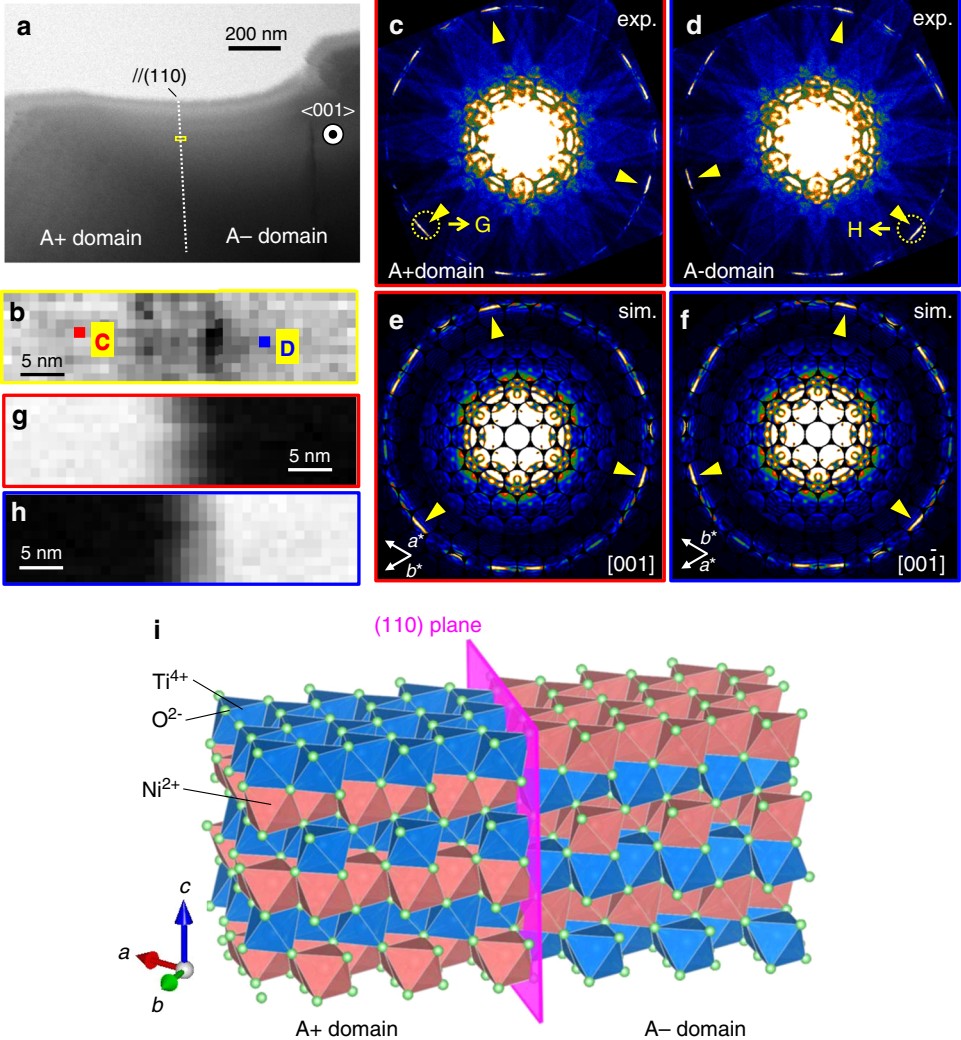

**Fig. 3 STEM-CBED maps and CBED patterns in the $R\bar{3}$ ferroaxial phase of NiTiO₃. a** Bright-field scanning transmission electron microscope (BF-STEM) image obtained with the 001 incidence, where a domain boundary // (110) is indicated by a white dotted line. Crystal orientations determined from the STEM-CBED patterns of (**c**) and (**d**) are schematically shown. **b** Magnified view of a yellow-box area in (**a**). **c, d** (exp.) Convergent-beam electron diffraction (CBED) patterns obtained with the 001 incidence at positions C and D shown in (**b**) are displayed in (**c**) and (**d**), respectively. These measurements were carried out at room temperature, that is, in the $R\bar{3}$ ferroaxial phase. **e, f** (sim.) Simulated CBED patterns of the $R\bar{3}$ phase of NiTiO₃ with the (**e**) [001] and (**f**) [00$\bar{1}$] incidence. The specimen thickness used for the simulations was 35 nm. The measured CBED patterns displayed in (**c**) and (**d**) well match up with the simulated (**e**) and (**f**), respectively [see higher-order Laue zone (HOLZ) reflections indicated by yellow arrowheads]. **g, h** STEM-CBED maps obtained from the intensities of the HOLZ reflections at G and H [yellow-dotted circles in (**e**) and (**f**), respectively]. **i** Orientations of A+ and A− domains and the domain boundary determined from the STEM-CBED measurements. Scale bar: 200 nm for (**a**) and 5 nm for panels (**b**), (**g**), and (**h**).

effect free from other electro-optical effects. More importantly, the sign of tensor component $\gamma_{333}$, which describes the situation when the directions of light propagation and an applied electric field are both parallel to a ferroaxial moment **A**, will depend on the sign of **A** (Supplementary Note 2). This means that the direction of $E$-induced optical rotation in A+ domain is opposite to that in A− domain. Therefore, ferroaxial domains can be distinguished by using the linear EG effect, which has been proposed in ref. [3].

As an indicator of linear EG effect, we use the coefficient $\alpha$ which relates the rotation angle of the light polarization plane $\phi$ to an applied voltage $V$. In general, optical rotatory power $\rho$ is given by

$$\rho = \tfrac{\pi}{\lambda n} g_{ij} l_i l_j. \qquad (2)$$

Here $l_i$ and $l_j$ are direction cosines of the wave normal, $n$ is the refractive index, $\lambda$ is the wavelength of the incident light, and the

Einstein notation is adopted. Furthermore, when the directions of light propagation and electric field are both parallel to **A**, $\phi$ (=$\rho d$ where $d$ is the sample thickness) is given by

$$\phi = \tfrac{\pi d}{\lambda n} \gamma_{333} E_3 l_3 l_3 = \tfrac{\pi}{\lambda n} \gamma_{333} V_3 , \qquad (3)$$

where $l_3 = 1$ and $V = E/d$. Therefore, $\phi$ is proportional to $V$ at fixed $\lambda$ and can be expressed as

$$\phi[\deg] = \alpha[\deg\, V^{-1}] \times V[V], \qquad (4)$$

in which the coefficient $\alpha$ ($\propto \gamma_{333}$) represents the magnitude of the linear EG effect.

Because the magnitude of the linear EG effect is usually small ($\alpha \leq 10^{-4}$ deg V$^{-1}$)[20], spatial distributions of EG have never been reported to date. To spatially resolve such small EG signals, we adopted a difference image-sensing technique which was recently developed for ferroelectrics field modulation imaging[24,25]. In this technique, microscopy images of transmitted light were captured

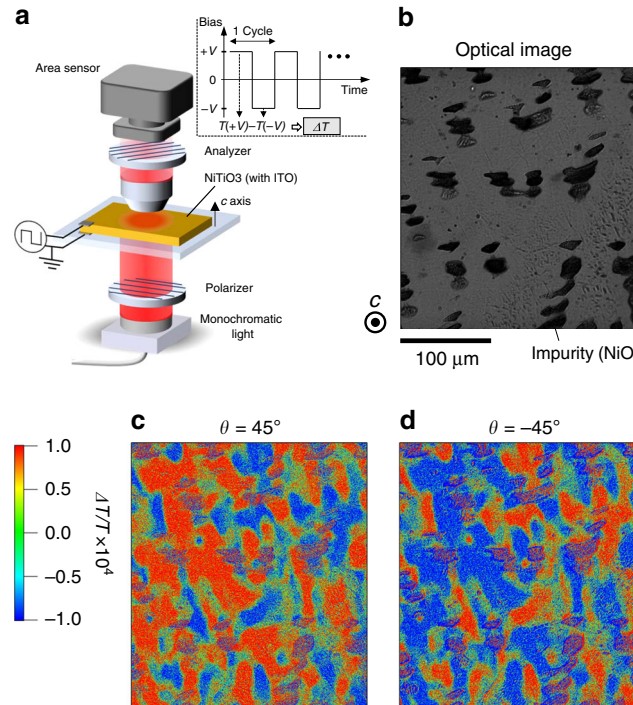

**Fig. 4 Spatial distribution of ferroaxial domains obtained via electrogyration in NiTiO₃. a** Experimental setup of electrogyration measurement using a difference image-sensing technique. Inset of (**a**) shows temporal evolution of applied voltage $V$ during the measurement. Microscopy images of transmitted light were captured by the area sensor while the positive and the negative $V$ applied. The difference of transmittance between the positive- and negative-voltage images ($\Delta T$) divided by the average of them ($T$) was calculated for each pixel detection, and then spatial distributions of $\Delta T/T$ were obtained. **b** Transmission optical microscopy image with the incidence of light along the $c$ axis (Scale bar: 100 μm). Dark areas in the image correspond to NiO impurity. **c, d** The two-dimensional maps of $\Delta T/T$, which corresponds to electrogyration, at the same area as panel **b**. A 3 × 3 median filter was applied to the raw images. The polarization direction of the analyzer was set at (**c**) $\theta = +45°$ and (**d**) $-45°$ with respect to that of the polarizer. These measurements were done under $V = \pm100$ V at room temperature, that is, in the $R\bar{3}$ ferroaxial phase. A $\Delta T/T$ color scale is applied to the images in panels **c** and **d**. Red and blue regions correspond to either A+ or A− ferroaxial domains. Purple-colored regions represent areas of NiO impurity.

by an area-image sensor while positive and negative voltages ($V$) applied. The difference of transmittance between the positive- and negative-voltage images ($\Delta T$) divided by the average of them ($T$) was calculated for each pixel detection, and then spatial distributions of $\Delta T/T$ were obtained. A schematic of the experimental setup is shown in Fig. 4a, and the measurement details are given in Methods. As described in Supplementary Note 3, $\Delta T/T$ is proportional to $\alpha$ representing the linear EG effect when the angle between the orientation of a polarizer and an analyzer ($\theta$) is set at $\theta = \pm45°$. The sign of $\theta$ is defined as positive when the polarization direction of the analyzer rotates clockwise with respect to that of the polarizer from the observer's point of view. The validity of this technique was confirmed by measurements of the linear EG effect in a reference material PbWO₄ (see Supplementary Note 4).

### Optical imaging of ferroaxial domains in NiTiO₃ via EG effect.
We examined ferroaxial domains of NiTiO₃ with the above-mentioned optical technique using the EG effect. The directions

of light propagation and an applied electric field were both parallel to the $c$ axis, meaning that EG corresponding to the $\gamma_{333}$ component was probed. Figure 4b displays the transmission optical microscopy image of the specimen used for the EG measurement with the incidence of light along the $c$ axis. In the image, there are dark island-shaped inclusions identified as NiO impurities by the energy dispersive X-ray analysis (Supplementary Note 5). Spatial distributions of $\Delta T/T$ at the same area as Fig. 4b were obtained under the applied voltage of $\pm100$ V in the polarization configurations at $\theta = \pm45°$. The results for $\theta = +45°$ and $-45°$ are displayed in Fig. 4c, d, respectively, in which red (blue) color corresponds to a positive (negative) sign of $\Delta T/T$. Note that the regions of NiO impurities (dark areas in Fig. 4b) appear purple (a mixture of red and blue) in Fig. 4c, d because the intensity of transmitted light in the region is too small to get meaningful signals. Except for the impurity regions, the images of Fig. 4c, d show a complete reversal of the contrast within the margin of error. This means that the observed $\Delta T/T$ is due to electric-field-induced change in optical rotation, i.e., EG, but not to that in optical absorption (see Supplementary Note 3). Therefore, red and blue regions in Fig. 4c, d correspond to either A+ and A− ferroaxial domains, and the color contrasts of these figures reflect the ferroaxial domain pattern in NiTiO₃.

To check whether EG observed in NiTiO₃ is ascribed to the linear effect and/or higher-order ones, we carried out measurements of the EG spatial distributions as a function of applied voltage $V$. Figure 5b–e shows spatial distributions of $\Delta T/T$ obtained in selected applied voltages at $\theta = +45°$ (b-d) and $-45°$ (f). The data were taken at a slightly different area from that of Fig. 4b–d. The color contrasts monotonically increase with increasing the magnitude of $V$ (Fig. 5b–d), and the contrasts get reversed by switching $\theta$ from $+45°$ to $-45°$ (compare Fig. 5d, e). We calculated the average of $\Delta T/T$ in the pixels at selected single ferroaxial domain areas (both red and blue) denoted by boxes in Fig. 5a–d and took its $V$ dependence. As seen in Fig. 5f, the magnitude of $\Delta T/T$, i.e., the magnitude of EG, is proportional to $V$. These results confirm that the electric-field-induced change in $\Delta T/T$ observed in NiTiO₃ is ascribed to the linear EG effect. We also calculated the magnitude of EG using the average of $\Delta T/T$ of the areas denoted by black and white boxes in Fig. 5d ($\pm100$ V, $\theta = +45°$), and obtained $\alpha = (2.0 \pm 1.0) \times 10^{-5}$ deg V$^{-1}$ for the red area and $(-1.9 \pm 0.9) \times 10^{-5}$ deg V$^{-1}$ for the blue area. The errors were calculated from the standard deviation of $\Delta T/T$.

The domain structures obtained by the optical imaging are irregular in shape (Fig. 4c, d). Furthermore, not only sharp domain boundaries but also relatively thick ones are present (see green areas between red and blue areas in Fig. 4c, d). This is also true for the results of CBED. In addition to the (110)-type sharp domain boundaries shown in Fig. 3, relatively thick boundaries where the two domains (A+ and A−) overlap were also obtained in our CBED measurement (see Supplementary Note 7). Thus, the result of the optical imaging is compatible with that of STEM-CBED. Furthermore, the length scales of the ferroaxial domains observed in NiTiO₃ are on the orders of $10^0 \sim 10^2$ μm. Such length scales are roughly comparable with the result of a previous SHG study which reported uneven domain populations in ferroaxial RbFe(MoO₄)₂ obtained from measurements using incident light with a 50-μm diameter spot on the sample[8].

### Discussion
Domains of ferroic materials are of technological importance because they have been widely used for information storage and for electric, magnetic, and optical switches. Besides conventional ferroic materials such as ferromagnetics, vector-like forms of ferroic orders, for example, ferro-toroidal order, attract much

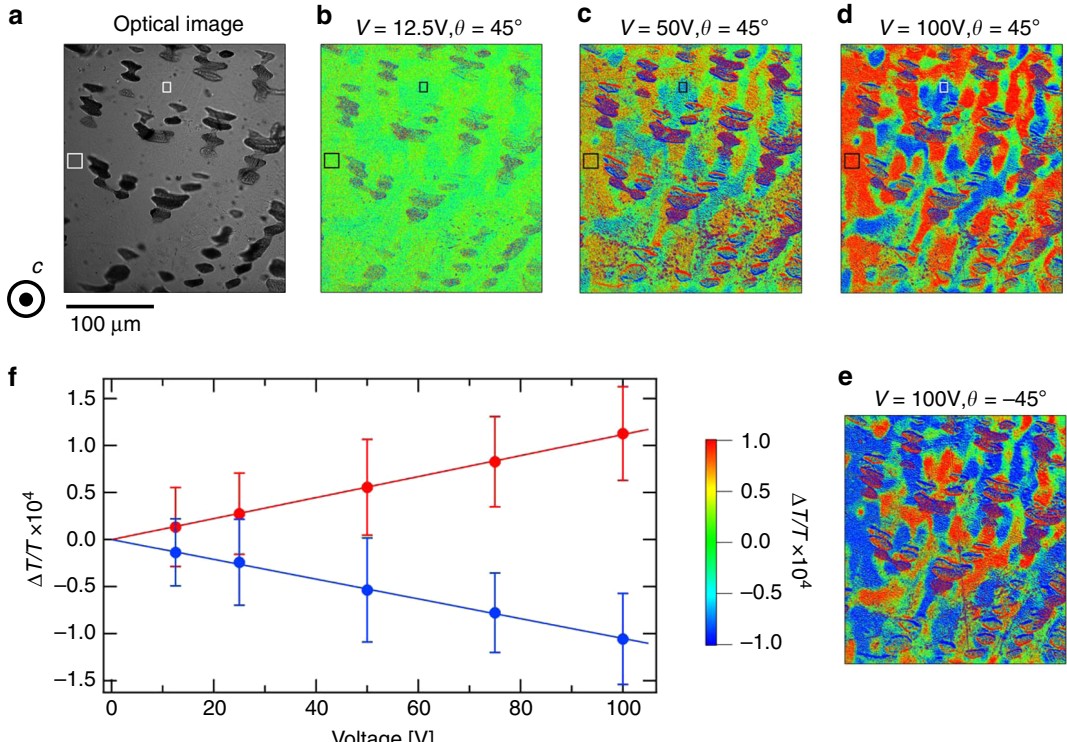

**Fig. 5 Applied voltage dependence of the intensity map of electrogyration in NiTiO₃. a** Transmission optical microscopy image with the incidence of unpolarized light along the $c$ axis (Scale bar: 100 μm). Dark areas in the image correspond to NiO impurity. **b–e** The two-dimensional maps of electrogyration at the same area as (**a**). To obtain the maps, the difference of the transmission microscope images at the positive and the negative voltages divided by the average of them ($\Delta T/T$) was calculated for each pixel detection. A 3 × 3 median filter was applied to the raw images. The polarization configuration was set at (**b–d**) $\theta = +45°$ and (**e**) $-45°$. The applied voltage $V$ was (**b**) ±12.5 V, (**c**) ±50 V, and (**d, e**) ±100 V. A $\Delta T/T$ color scale is applied to the images in (**b–e**). **f** The $V$ dependence of the average of $\Delta T/T$ taken at $\theta = +45°$ in the selected single domain areas denoted by boxes (**a–d**). The red and blue dots correspond to the data of the areas surrounded by large and small boxes, respectively, in each panel. The standard deviation is shown as an error bar. The lines denote least squares fitting lines.

interests in the past decades[3,26,27]. Furthermore, the recent observation of ferroaxial (or ferro-rotational) order in displacive type RbFe(MoO₄)₂ using rotational anisotropy SHG[8] has added another ferroic ordered state with vector-like order parameters. The present study reveals spatial distributions of ferroaxial domains in NiTiO₃ on both nanometer and micrometer scales by means of STEM-CBED and linear optical microscopy using the EG effect. Our achievement is not only the visualization of ferroaxial domains formed in a material undergoing a pure ferroaxial transition but also the verification of an order–disorder type ferroaxial crystal. The ferroaxial transition in NiTiO₃ studied here corresponds to a transition from the disordered corundum structure to the ordered ilmenite structure. Indeed, such a transition has been extensively studied in FeTiO₃-Fe₂O₃ solid solution series[28–30]. From the view point of those days, domains formed due to the ordered-disordered transition were recognized as antiphase domains (or merohedral twin domains). However, from a modern perspective, such a transition can be also classified into ferroaxial order. Thus, the present study will also link these former studies with the current growing interest in ferroaxial materials.

To manipulate the ferroaxial domain states in pure ferroaxial crystals free from ferroelectric or ferroelastic order, the combination of an electric field and a stress is expected to be effective from the symmetry-based consideration[9]. At the moment, however, the nature of the solitary conjugate field for ferroaxial order remains unsolved and the manipulation of the domain states has never been demonstrated experimentally. Our success of the domain visualization will provide intriguing opportunities to

tackle such unsolved problems of ferroaxial materials. Furthermore, the ferroaxial order is closely related to other ferroic orders and they are sometimes coupled. Therefore, the study of ferroaxial order will lead to new multifunctional materials.

## Methods

**Sample preparation and characterization.** Single crystals of NiTiO₃ were prepared by the floating zone method[31]. First, polycrystalline feed rods were prepared by a solid-state reaction. Powders of NiO and TiO₂ with 99.9% purity were weighted to the prescribed ratios, mixed, well grounded, and heated at 1000 °C for 10 h in air. The resulting polycrystalline samples were ground into powders again and then pressed into rods with a dimension of about 6 mm in diameter and 100 mm in length. The rods were sintered again at 1100 °C for 15 h in air. The crystal growth was carried out on the sintered rods with the use of a halogen-lamp image furnace at a feed rate of 1.0 mm/h in flowing air. As a result, yellowish-brown crystals were obtained. Powder x-ray diffraction (XRD) and scanning electron microscopy—energy-dispersive x-ray spectroscopy (SEM-EDX) revealed that the obtained crystals mainly consist of the ilmenite-type NiTiO₃ phase but include a small amount of NiO and TiO₂ impurities (Supplementary Note 5). For measurements of EG, one of the crystals was oriented by using Laue XRD patterns, cut into a thin plate shape with the widest faces normal to the $c$ axis (hexagonal setting), and polished down to the thickness of about 60 μm. To form transparent electrodes that allow the application of a voltage parallel to the $c$ axis, indium/tin-oxide (ITO) was spattered onto the widest faces.

**STEM-CBED measurements.** Measurements of scanning transmission electron microscopy and convergent-beam electron diffraction (STEM-CBED) were conducted using a JEM-2010FEF transmission electron microscope at an accelerating voltage of 100 kV. The microscope is equipped with a Schottky-type field emission gun, an Omega-type energy-filter, and a STEM unit with a Gatan STEM diffraction imaging system[15,16]. Specimens for the STEM-CBED measurements were prepared by Ar ion milling of a NiTiO₃ crystal. The obtained fragments were dispersed onto microgrids for electron microscopy. CBED patterns of the specimens were taken at room temperature by using a 4k × 4k CMOS camera (Gatan Rio camera) and the

STEM diffraction imaging system. To obtain STEM-CBED maps, the convergent-beam electron probe was scanned with a step of 1 nm and exposure time of 0.5 s for a single CBED pattern. The electron probe size was ~1 nm in diameter. Computer simulations of the CBED patterns for the structure model of $NiTiO_3$ were also performed using the software MBFIT[32,33] based on the Bloch-wave dynamical theory of electron diffraction. The lattice parameters and the atom positions were taken from the room-temperature data reported in ref. [34]. The atomic scattering factors for the ions of $Ni^{2+}$, $Ti^{4+}$, and $O^{2-}$ were used.

**Imaging of ferroaxial domains via EG effect.** Two-dimensional maps of ferroaxial domains were obtained using a polarized-light microscope in the transmittance geometry, as illustrated in Fig. 4a. A light-emitting diode (LED) whose wavelength is 660 nm was used as light source. This is because the absorbance around this wavelength is relatively low in $NiTiO_3$[35] and then a sufficient amount of transmitted light can be obtained. Microscopy images of transmitted light were captured by a CMOS-area-image sensor. To measure optical rotation in the presence of an electric field, the $NiTiO_3$ specimen with ITO transparent electrodes was placed between a polarizer and an analyzer. Both the directions of light propagation and the electric field were along the $c$ axis of the specimen. The polarization direction of the analyzer was set at ±45° with respect to that of the polarizer. To detect small EG signals, i.e., electric-field-induced change in optical rotation, we adopted a difference image-sensing technique[24,25]. A square-wave bias voltage (up to ±100 V) was applied between the electrodes at a frequency of 20 Hz. Pulsed light at a frequency of 40 Hz from the LED was irradiated onto the specimen, where the pulsed irradiation was synchronized with the applied square-wave voltage. Microscopy images of transmitted light were captured by the CMOS-area-image sensor while positive and negative voltages applied. To suppress the noise, 16,384 images are captured for the positive- and negative-voltage states and are averaged. The difference between the positive- and negative-voltage images divided by the average of them ($\Delta T/T$) was calculated for each pixel detection, and the spatial distribution of $\Delta T/T$ was obtained. Also, a $3 \times 3$ median filter was applied to the raw images. The derivation of the EG from $\Delta T/T$ is described in Supplementary Note 3. The validity of this technique for EG measurements was confirmed by comparing the results obtained with this technique and those obtained with a lock-in technique (Supplementary Note 6) on a reference material, $PbWO_4$ (Supplementary Note 4).

## Data availability
The data that support the findings of this study are available from the corresponding author upon reasonable request.

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

## Acknowledgements
We thank D. Hamane for his help with sample preparation for optical measurements and sample characterization by SEM-EDX measurements. These experiments were carried out under the Visiting Researcher's Program of the Institute for Solid State Physics, the University of Tokyo. We also thank M. Nagai for his advice about electrogyration experiments and M. Ageishi for his help with sample preparation for TEM experiment. The images of crystal structures were drawn using the software VESTA[36]. This work was partially supported by JSPS KAKENHI (Grant Numbers JP17H01143, JP19H05823, JP18H03674, JP18K18931, JP19K14623, and JP19H01847), the MEXT Leading Initiative for Excellent Young Researchers (LEADER), and JST CREST Grant Number JPMJCR18J2.

## Author contributions

T.H., K.K., and T.K. conceived the project. With the help of K.K., T.H. performed crystal growth, sample preparation, and electrogyration measurements by using a lock-in technique. S.H. prepared transparent electrodes for electrogyration measurements. T.H., Y.U., S.M., and T.H. carried out optical imaging measurements. D.M. and K.T. performed STEM-CBED measurements and simulations. T.H. and T.K. prepared the manuscript, and all authors discussed and contributed to the final manuscript.

## Competing interests

The authors declare no competing interests.
