## [Peer Review File · Nature Communications]

REVIEWER COMMENTS

Reviewer #1 (Remarks to the Author):

In this manuscript, T. Hayashida et al used a combination of scanning transmission electron microscopy-convergent beam electron diffraction (STEM-CBED) and linear electrogyration (EG) optical microscope to probe the presence of ferroaxial domains in NiTiO₃ at both nanoscale and micron-scale. The experiments were carried out carefully and the analysis was done rigorously. Despite the high-quality of the manuscript in its present form, the following questions need to be addressed before recommendation for publication.

1. Looking at the diagrams for A⁻ and A⁺ domains in Figure 2(d) and (e), it seems that both of them have three vertical mirrors at 0°, 60°, and 120° from the axis a, and that the two domains can be related by a 180° rotation about c axis. These two observations above are not consistent with what the two domains of ferroaxial order should show. Could the authors please explain first and then improve the diagrams? This same question applies to Figure 3(j).

2. Two different CBED patterns were observed in Figure 3(c) and (d) that are related to each other by a “mirror image” (line 99 of main text). Can the authors please specify the mirror direction with respect to crystal axes a and b? In addition, can the authors please compare this mirror direction to the mirrors present in the high-symmetry phase of point group $\bar{3}m$ (space group $R\bar{3}c$)? This information is important here for justifying that the two CBED patterns are from the two ferroaxial domains with opposite ferroaxial vectors, because the two domains can only be related by the mirrors originally present in the high-symmetry phase.

3. Looking at the linear EG images in Figure 4(c) and (d), as well as Figure 5(d) and (e), there is a green colored barrier (with $\Delta T/T=0$) between every adjacent red (positive $\Delta T/T$) and blue (negative $\Delta T/T$) domains. The estimated width for the green barriers is in order of 10 μm , significantly greater than the optical diffraction limit. Can the authors please explain what might cause these green barriers with zero $\Delta T/T$? Recall that the boundary between two domains in Figure 3(b), (g), and (h) is as sharp as a few nm. Can the authors please discuss the difference of the boundary characteristics between those detected by STEM and linear EG imaging?

4. In line 135 – 138 of main text, the authors established that $i=j=k=3$ when the light propagating direction and the DC electric field are parallel to the moment A. I understand the index j picks that of the optical wavevector (light propagating direction) whereas the index k selects that of the DC electric field. However, it is not quite clear for me how the index i is related to the direction of A. Can the authors please elaborate on this point?

5. In line 138 of main text, what does “(2 for 2/m)” mean?

6. In line 51 of SI, the authors used a mirror to show that the sign of moment A is encoded in the axial tensor γ_{ijk} . Can the authors clarify if this mirror here corresponds to one of three mirrors present in the $R\bar{3}c$ phase?

Reviewer #2 (Remarks to the Author):

This paper presents the first electrogyration imaging of a ferroaxial crystal undergoing a pure ferroaxial phase transition. The results are confirmed with STEM/CBED.

It is a very clear and simple experiment, but well thought out. Since electrogyration may get mixed up with electro-optic and inverse piezoelectric effects, the authors have been careful to pick a material symmetry group that does not have this effect. They do show a linear dependence of gyration with electric field.

Overall a nice elegant study!

A couple of comments:

(1) the tensor components being probed here for electrogyration is not explicitly described. I would recommend doing so.

(2) In the intro: The usage of torroidal order is inaccurate (and it is unfortunate in literature). A torus is not simply $r \times p$. A torus is a surface of revolution generated by revolving a circle in three-dimensional space about an axis that is coplanar with the circle (see wiki page). This might at least make some sense if p was an axial vector, but not if p was just electro polarization. $r \times p$ would simply be a vortex. The authors have an opportunity to clarify or at least not muddy the field even more by repeating literature usage.

Reviewer #3 (Remarks to the Author):

The paper by Hayashida et al. reports the visualization of ferroaxial domains using linear electrogyration effect as well as the STEM-CBED method in NiTiO₃. These measurements give convincing evidence of the spatial distributions of ferroaxial domains in NiTiO₃ on both nanometer and micrometer scales. Their data are of high quality, and this paper is the very 1st report on ferroaxial domains using an optical technique. Emphasize, that as authors noted, space inversion and time reversal symmetries are not broken in ferroaxial systems, so it is not straightforward to visualize ferroaxial domains, at least in optical methods. Thus, little studies have been performed on ferroaxial systems even though there are plenty of ferroaxial systems. The possible visualization of ferroaxial domains using linear optical gyration was, for the first time, noted in Ref. 3: in its abstract, "In particular, the conditions for non-reciprocity of ferro-rotational order are discussed and the possible use of linear optical gyration is suggested as a way to detect ferro-rotational domains." is stated. (Ferro-rotational is identical with ferroaxial.) Thus, it is fair to state this prediction clearly in this manuscript. The big family having hematite to ilmenite (R-3c to R-3) order-disorder transition (also ferroaxial transition) has been known and been investigated extensively including the microstructures using dark-field TEM. One example is: Order-disorder transition-induced twin domains and magnetic properties in ilmenite-hematite Gordon L. Nord and C. A. Lawson, American Mineralogist, Volume 74, pages 160-176, 1989. These earlier papers need to be cited.

Response to the reviewers' comments

We would like to sincerely thank all the reviewers for their careful reading of our manuscript as well as for constructive comments to improve the quality of the manuscript. In the following, we reply to the respective reviewers' comments.

Hereinafter, the comments from each reviewer appear in black italic letters while our reply is written with blue letters. Significant revisions and newly added descriptions are indicated with red letters in the manuscript.

[Response to Reviewer #1]

(1-1) *Looking at the diagrams for A- and A+ domains in Figure 2(d) and (e), it seems that both of them have three vertical mirrors at 0° , 60° , and 120° from the axis a , and that the two domains can be related by a 180° rotation about c axis. These two observations above are not consistent with what the two domains of ferroaxial order should show. Could the authors please explain first and then improve the diagrams? This same question applies to Figure 3(j).*

We agree with the reviewer that the diagrams seem to have three vertical mirrors and that the two domains can be related by the two-fold rotation about the c axis. However, this is not the case. To explain this discrepancy, we show in Fig. R1 the arrangement of Ti ions within a single TiO_6 layer. As seen in Fig. R1a, the projection view along the c axis seems to have three vertical mirrors (brown dashed line). However, the z coordinates between the nearest neighboring Ti ions are different, which is explicitly depicted in the projection view from the a axis (Fig. R1b). From this figure, it is obvious that no vertical mirror is present at 0° from the a axis. In addition, the 180° rotation about the c axis causes the change of the z coordinates of the respective Ti ions. Thus, A+ and A- domains cannot be related by a 180° rotation about the c axis.

Fig. R1

In Figs. 2d,e, we intended to show the difference of A- and A+ domains in terms of the sign of rotational distortions. However, we learned from the reviewer's comment that the diagrams confuse readers. Therefore, responding to the reviewer's suggestion, we modify the diagrams of Fig. 2 in the revised manuscript. In the revised Figs. 2d,e, we simplify the diagram and pick only two Ti ions [$\text{Ti1}^{(*)}$ and $\text{Ti2}^{(*)}$] and six oxygen ions. These ions form two TiO_3 triangular pyramids which are related by the space inversion with the inversion center at the midpoint between Ti1 and Ti2 ions. The A+ and A- domains are related to each other by the

operations whose symmetries are lost at the ferroaxial transition {e.g., two-fold rotation about [110] and *c*-glide operation with glide plane || (110)}. In the revised Fig. 2d,e, red arrows denote the direction of (virtual) displacements of oxygen ions from the (110)-type planes (dotted lines in the figure) that correspond to the average oxygen positions between A⁺ and A⁻. In order to avoid misleading readers, we delete the diagram shown in Fig. 3j.

Related to the revisions of the figures, we revise/add the figure captions and some sentences in the subsection “Order-disorder type ferroaxial transition in NiTiO₃” on p. 3. The revised and newly added parts are written with red letters in the revised manuscript. We thank the reviewer for the constructive comment.

(1-2) Two different CBED patterns were observed in Figure 3(c) and (d) that are related to each other by a “mirror image” (line 99 of main text). Can the authors please specify the mirror direction with respect to crystal axes a and b? In addition, can the authors please compare this mirror direction to the mirrors present in the high-symmetry phase of point group $\bar{3}m$ (space group $R\bar{3}c$)? This information is important here for justifying that the two CBED patterns are from the two ferroaxial domains with opposite ferroaxial vectors, because the two domains can only be related by the mirrors originally present in the high-symmetry phase.

We confirm that the mirror plane relating the CBED pattern of Fig. 3c to that of Fig. 3d is parallel to the (110) plane, and revise/add the following sentences at the end of the first paragraph of the subsection “**Identification of ferroaxial domains by STEM-CBED measurement**” on p. 3.

“...indicating that the CBED patterns of Figs. 3c and 3d are almost in a mirror image relation whose mirror plane is parallel to (110). Note that such a mirror operation is one of the symmetry elements which are present in the high-temperature $\bar{3}m$ phase but lost in the low-temperature $\bar{3}$ phase.”

(1-3) Looking at the linear EG images in Figure 4(c) and (d), as well as Figure 5(d) and (e), there is a green colored barrier (with $\Delta T/T=0$) between every adjacent red (positive $\Delta T/T$) and blue (negative $\Delta T/T$) domains. The estimated width for the green barriers is in order of 10 μm , significantly greater than the optical diffraction limit. Can the authors please explain what might cause these green barriers with zero $\Delta T/T$? Recall that the boundary between two domains in Figure 3(b), (g), and (h) is as sharp as a few nm. Can the authors please discuss the difference of the boundary characteristics between those detected by STEM and linear EG imaging?

As pointed out by the reviewer, not only sharp domain boundaries but also relatively thick ones (green colored barriers having the width in order of 10⁰ μm) are present in the linear EG images (Figs. 4c,d). To reply the reviewer’s comment, we carried out additional measurements of CBED patterns at various sample positions. As a result, we also observed relatively thick domain boundaries by the CBED measurements. To show the new experimental result, we add Supplementary Note 7 which displays a BF-TEM image of the

area including a thick domain boundary (green colored area in Fig. S5). The CBED patterns obtained in the green colored region of Fig. S5 showed the superposition of A+ and A- domains' patterns. Therefore, the green barriers with zero $\Delta T/T$ in the EG images is likely to correspond to the areas in which A+ and A- domains coexist in the length scale below the resolution limit. A possible interpretation for the observed feature is that the domain boundary is NOT parallel to the c axis but considerably tilted from the c axis. Anyway, various length scales of domain boundaries were observed in both the experimental techniques.

Related to this revision, we changed the 1st part of the last paragraph in the subsection “**Optical imaging of ferroaxial domains in NiTiO₃ via electrogyration effect**” (p.6), as follows.

We add the following sentence.

“Furthermore, not only sharp domain boundaries but also relatively thick ones are present (see green areas between red and blue areas in Figs. 4c,d). This is also true for the results of CBED. In addition to the (110)-type sharp domain boundaries shown in Fig. 3, relatively thick boundaries where the two domains (A+ and A-) overlap were also obtained in our CBED measurement (see Supplementary Note 7).”

We removed the following sentence.

~~“However, only the combination of the three (110)-type domain boundaries [(110), (120), or (210)] can make macroscopic domains arbitrary in shape.”~~

(1-4) In line 135 – 138 of main text, the authors established that $i=j=k=3$ when the light propagating direction and the DC electric field are parallel to the moment A . I understand the index j picks that of the optical wavevector (light propagating direction) whereas the index k selects that of the DC electric field. However, it is not quite clear for me how the index i is related to the direction of A . Can the authors please elaborate on this point?

In the absence of natural optical rotation, optical rotatory power induced by an external electric field \mathbf{E} is given by [c.f. Nye, J. F., *Physical Properties of Crystals* pp.260-274 (Oxford Clarendon Press, New York, 1957)]

$$\rho = \frac{\pi}{\lambda n} g_{ij} l_i l_j = \frac{\pi}{\lambda n} \gamma_{ijk} E_k l_i l_j,$$

where l_i and l_j are direction cosines of the wave normal of incident light and the Einstein notation is adopted (take the summation about i , j and k). In the previous manuscript, we used the indices of a_i and a_j instead of l_i and l_j , respectively. However, we consider that using the same alphabet "a" as that for the elements of transformation matrix of mirror operation given in Supplementary Note 2 confuses readers. Therefore, we replace a_i and a_j with l_i and l_j , respectively, in the revised manuscript.

The light propagating parallel to the moment \mathbf{A} , i.e., the light propagating along the principal axis is described as

$$\mathbf{l} = \begin{pmatrix} 0 \\ 0 \\ 1 \end{pmatrix},$$

where the principal axis is taken as the z axis. Therefore, optical rotatory power in this direction is given by

$$\rho = \frac{\pi}{\lambda n} g_{33} l_3 l_3 = \frac{\pi}{\lambda n} \gamma_{333} E_3 l_3 l_3 = \frac{\pi}{\lambda n} \gamma_{333} E_3.$$

Thus, both of the indices of i and j are related to the direction of \mathbf{A} .

Thanks to the reviewer's question, we noticed that our descriptions about electrogyration is not clear enough. Therefore we modify the definition of ϕ in the 2nd paragraph of the subsection **"Electrogyration as a tool to observe ferroaxial domains"** (p.5), and limit the situation to the configuration that **the directions of light propagation and electric field are both parallel to \mathbf{A} .**

(1-5) In line 138 of main text, what does "(2 for 2/m)" mean?

In the case of the monoclinic system, the y axis is often taken as the principal axis (// two-fold axis). Therefore, we wanted to say " $i = j = k = 2$ for point group $2/m$ ". However, we find this description confuses readers. Therefore, we add the following sentence in the first paragraph of the subsection **"Electrogyration as a tool to observe ferroaxial domains"** on p. 4.

"Hereinafter, the z axis (the third axis) is taken as the principal axis."

In addition, the corresponding sentence at the end of p.4 is revised as follows.

"More importantly, the sign of the tensor component γ_{333} , that describes the situation when the directions of light propagation and an applied electric field are both parallel to a ferroaxial moment \mathbf{A} , will depend on the sign of \mathbf{A} (Supplementary Note 2)."

(1-6) In line 51 of SI, the authors used a mirror to show that the sign of moment A is encoded in the axial tensor γ_{ijk} . Can the authors clarify if this mirror here corresponds to one of three mirrors present in the $R\bar{3}c$ phase?

We intended to use the (110) mirror plane that is one of the three mirrors present in $\bar{3}m$. However, we noticed that the transformation matrix of σ_v in the previous manuscript was incorrect. The correct one is

$$\sigma_v = \begin{bmatrix} a_{11} & a_{12} & a_{13} \\ a_{21} & a_{22} & a_{23} \\ a_{31} & a_{32} & a_{33} \end{bmatrix} = \begin{bmatrix} \frac{1}{2} & -\frac{\sqrt{3}}{2} & 0 \\ -\frac{\sqrt{3}}{2} & -\frac{1}{2} & 0 \\ 0 & 0 & 1 \end{bmatrix}$$

in the orthogonal basis. This correction does not affect the calculation of γ_{333} because only $a_{33} = 1$ which is unchanged contributes to the calculation. We revise the transformation matrix in Supplementary Note 2 and add the following sentence just above the matrix.

“The orthogonal basis (\mathbf{a}_o , \mathbf{b}_o , and \mathbf{c}_o) is obtained from the hexagonal basis (\mathbf{a}_h , \mathbf{b}_h , and \mathbf{c}_h) according to the relations, $\mathbf{a}_o \parallel \mathbf{a}_h$, $\mathbf{c}_o \parallel \mathbf{c}_h$, and $\mathbf{b}_o \parallel \mathbf{a}_h \times \mathbf{c}_h$. For a mirror operation [mirror plane $\parallel (110)$]”

[Response to Reviewer #2]

(2-1) *the tensor components being probed here for electrogyration is not explicitly described. I would recommend doing so.*

Responding to this comment, we add the following sentence after the 1st sentence of the subsection “**Optical imaging of ferroaxial domains in NiTiO₃ via electrogyration effect**” (p.5) to explicitly describe the tensor component probed in our experiment.

“The directions of light propagation and an applied electric field were both parallel to the c axis, meaning that EG corresponding to the γ_{333} component was probed.”

(2-2) *In the intro: The usage of toroidal order is inaccurate (and it is unfortunate in literature). A torus is not simply $r \times p$. A torus is a surface of revolution generated by revolving a circle in three-dimensional space about an axis that is coplanar with the circle (see wiki page). This might at least make some sense if p was an axial vector, but not if p was just electro polarization. rxp would simply be a vortex. The authors have an opportunity to clarify or at least not muddy the field even more by repeating literature usage.*

We thank the reviewer for pointing out our misunderstanding about the usage of the term “toroidal”. We understand that the term “electric toroidal moment” is not appropriate for the moment \mathbf{A} formed by “ $\mathbf{r} \times \mathbf{p}$ ”. Except for “electric toroidal moment”, several terms were used in former literatures to express \mathbf{A} [e.g., “ferro-rotation moment” (ref. 3), “axial vector” (refs. 4-6), and “rotational moment” (ref. 8)]. In the present paper, we use the term “ferroaxial” rather than “ferro-rotational”. Therefore, we adopt the term “ferroaxial moment”, and replace “electric toroidal moment” with “ferroaxial moment (or ferro-rotation moment)” in the 1st sentence of introduction section (p. 1) and the caption of Fig. 1. We also remove the description “electric ferro-toroidal order” at the end of p. 1.

[Response to Reviewer #3]

(3-1) *The possible visualization of ferroaxial domains using linear optical gyration was, for the first time, noted in Ref. 3: in its abstract, “In particular, the conditions for non-reciprocity of ferro-rotational order are discussed and the possible use of linear optical gyration is suggested as a way to detect ferro-rotational domains.” is stated. (Ferro-rotational is identical with ferroaxial.) Thus, it is fair to state this prediction clearly in this manuscript.*

Following the reviewer’s suggestion, we cited Ref. 3 by revising the following sentence at the last sentence of the 1st paragraph in the subsection “**Electrogyration as a tool to observe ferroaxial domains**” (p. 4).

“Therefore, ferroaxial domains can be distinguished by using the linear EG effect, which has been proposed in ref.³.”

(3-2) The big family having hematite to ilmenite (R-3c to R-3) order-disorder transition (also ferroaxial transition) has been known and been investigated extensively including the microstructures using dark-field TEM. One example is: Order-disorder transition-induced twin domains and magnetic properties in ilmenite-hematite Gordon L. Nord and C. A. Lawson, *American Mineralogist*, Volume 74, pages 160-176, 1989. These earlier papers need to be cited.

Thanks to this reviewer's comment, we could notice the previous studies closely related to the present study. We agree with the reviewer that these earlier papers need to be cited. Therefore, we add the following sentences at the end of the 1st paragraph of "**Outlook**" section and cite several earlier papers.

"The ferroaxial transition in NiTiO₃ studied here corresponds to a transition from the disordered corundum structure to the ordered ilmenite structure. Indeed, such a transition has been extensively studied in FeTiO₃-Fe₂O₃ solid solution series²⁷⁻²⁹. From the view point of those days, domains formed due to the ordered-disordered transition were recognized as antiphase domains (or merohedral twin domains). However, from a modern perspective, such a transition can be also classified into ferroaxial order. Thus, the present study will also link these former studies with the current growing interest in ferroaxial materials."

The newly cited references are

27. Lawson, C. A., Nord, G. L., Dowty, E. & Hargraves, R. B. Antiphase domains and reverse thermoremanent magnetism in ilmenite-hematite minerals. *Science* **213**, 1372–1374 (1981).
28. Nord, G. L. & Lawson, C. A. Order-disorder transition-induced twin domains and magnetic properties in ilmenite-hematite. *Am. Mineral.* **74**, 160–176 (1989).
29. Nord, G. L. Transformation-induced twin boundaries in minerals. *Phase Transitions* **48**, 107–134 (1994).

Aside from reviewers' comments, we made the following changes.

In the previous manuscript, the "clockwise" rotational arrangement of dipoles was defined as "A+ domain" while the "anticlockwise" one was "A- domain". However, following the conventional setting of vector product, we reversed the signs, i.e., the anticlockwise arrangement as A+ and the clockwise arrangement as A- in the revised manuscript.

Also, following *Nature's* policy about Image integrity, we add the notation about the median filter, which was applied to the maps of $\Delta T/T$, in the legends of Fig. 4, Fig. 5, Fig. S2, and in the subsection "**Imaging of ferroaxial domains via electrogyration effect**" of the section "**Methods**".

These changes do not affect the discussion and the conclusion of this paper.

REVIEWERS' COMMENTS:

Reviewer #1 (Remarks to the Author):

The authors have answered all my questions clearly and thoroughly. I very much enjoyed reading this manuscript and appreciate the detailed explanation from the authors. Therefore, I recommend acceptance of this manuscript for publication at Nature Communications.

Reviewer #3 (Remarks to the Author):

The reply is good - I suggest to publish it as revised.